# Evaluation of d-Limonene and *β*-Ocimene as Attractants of *Aphytis melinus* (Hymenoptera: Aphelinidae), a Parasitoid of *Aonidiella aurantii* (Hemiptera: Diaspididae) on *Citrus* spp.

**DOI:** 10.3390/insects11010044

**Published:** 2020-01-08

**Authors:** Khalid Mohammed, Manjree Agarwal, Beibei Li, James Newman, Tao Liu, Yonglin Ren

**Affiliations:** 1College of Science, Health, Engineering and Education, Murdoch University, Perth, Western Australia 6150, Australia; k.mohammed@murdoch.edu.au (K.M.); m.agarwal@murdoch.edu.au (M.A.); james.newman@cbh.com.au (J.N.); 2College of Agriculture and Forestry, University of Mosul, Mosul 41002, Iraq; 3College of agriculture, Kansas State University, Manhattan, KS 66502, USA; libeibei@ksu.edu; 4Institute of Equipment Technology, Chinese Academy of Inspection and Quarantine, Beijing 100123, China

**Keywords:** California red scale, citrus fruits, volatile organic compounds, herbivore-induced, parasitoid, SPME-GC-MS

## Abstract

The volatile organic compounds (VOCs) released from herbivore-infested plants can be used as chemical signals by parasitoids during host location. In this research, we investigated the VOC chemical signals for the parasitoid *Aphytis melinus* to discriminate between *Aonidiella aurantii* (California red scale)-infested fruit and non-infested fruit on three different citrus species. First, we identified the chemical stimuli emanating from non-infested and *A. aurantii*-infested citrus fruits via solid phase microextraction (SPME) and gas chromatography-mass spectrometry (GC-MS) analyses and identified 34 volatile organic compounds (VOCs). The GC-MS analysis showed qualitative and quantitative differences between VOCs emitted from non-infested and infested citrus fruit. Two VOCs, d-limonene and *β*-ocimene, were significantly increased in all infested fruit, regardless of the fruit species. The response of the female adult *A. melinus* to olfactory cues associated with *A. aurantii* infested fruit was evaluated using a Y-tube olfactometer. In two-choice behavioural assays, *A. melinus* females preferred infested citrus cues over non-infested fruit. Females showed positive chemotaxis toward these VOCs in all tested combinations involving two dosages of synthetic compounds, d-limonene and *β*-ocimene, except for d-limonene at a dosage of 10 μL/mL. The application of these VOCs may help to enhance the effectiveness of bio-control programs and parasitoid mass-rearing techniques.

## 1. Introduction

The citrus industry is considered one of the largest fresh fruit industries in Australia, as well as the largest fresh fruit exporter, with an annual average export volume of 170,000 tonnes and a value of AUS$190 million [1]. *Aonidiella aurantii* (Maskell) (Hemiptera: Diaspididae), is a major insect pest of citrus crops, causing significant production losses [2]. *A. aurantii* prefers to attack citrus, and it is one of the most important citrus pests worldwide [3]. *A. aurantii* attacks all citrus species, but there are varying levels of susceptibility, so the order of vulnerability in descending is lemon trees (*Citrus limon* (L.) Osbeck), grapefruit (*C. xparadisi* Macf.), orange (*C. sinensis* (L.) Osbeck) and two different species of mandarin (*C. reticulata* Blanco and *C. unshiu* Markovitch) [4]. California red scale *A. aurantii* attacks all aerial parts of the citrus tree including fruits, leaves and branches and twigs [5]. However, the fruit is the preferred plant substrate for *A. aurantii*, followed by leaves, while branches and twigs are the least preferred substrate [6]. This is economically important as the presence of this scale on fruit considerably reduces their market value, causing huge economic losses [7]. Chemical control of *A. aurantii* is difficult and is frequently followed by reinfestation within a short period, resistance to different products used for its control, and the risk of eliminating natural enemies present in the field [8].

Plant VOCs can serve as semiochemicals to protect plants from insect and pathogen attack, attract beneficial animals, and as communication signals within and among plants [9]. Terpenoids and other VOCs emitted from plants in response to insect attack allow parasitoids and predators to distinguish between infested and non-infested plants and thus aid in locating hosts or prey [10]. In addition, the herbivore-induced plant volatiles (HIPVs) vary quantitatively and qualitatively related to various biotic and abiotic factors (e.g., number of infesting pests, period and duration of infestation and previous infestation) and the VOCs are specific to each plant–pest association [11]. It is well-known that insect feeding activity leads to biochemical changes in plants. Plants respond to the presence of pests by activating their defence system [12], but they can also trigger indirect defences, such as the emission of HIPVs [13]. The role of VOCs on host location by natural enemies has been widely investigated [14,15]. Many parasitoids of phytophagous insects orient to VOCs, using them as VOCs to guide their search for hosts [11,13] and differential emissions have been reported for *A. aurantii*-infested and non-infested fruits [16].

The aphelinid *A. melinus* is a primary ectoparasitoid of armoured scale insects. It is the most commonly used biocontrol agent for California red scale *A. aurantii* across the world, through augmentative releases [17,18]. *A. melinus* females rely on stimuli to successfully locate their host. *Aphytis* use learned volatile cues from host plants as long-range attractants to potential habitats of their hosts [19,20]. Previous work with *A. melinus* and California red scale showed that wasps were attracted to the odours of scale-infested lemon fruit [21].

In this study, we investigated the olfactory cues used by *A. melinus* females to locate their host microhabitat. We hypothesise that the VOCs from *A. aurantii* infested citrus fruit may play a pivotal role in affecting *A. melinus* host location. Three different citrus fruit, lemon fruit, orange fruit and mandarin fruit, were tested to determine parasitoid attractiveness and VOC emissions. First, the VOCs emitted from non-infested and infested citrus fruits were extracted with SPME fibre and analysed by gas chromatography-mass spectrometry to determine whether that VOCs are attributable to herbivores’ activity and determine possible VOCs. Subsequently, we evaluated female preferences for infested or non-infested fruit in olfactometer bioassays. Finally, to understand the magnitude of volatile attractiveness, the innate attractiveness of VOCs to *A. melinus* females in varying concentrations was tested in the laboratory.

## 2. Materials and Methods

### 2.1. Insect Colonies

Parasitoid pupae of *A. melinus,* purchased from Biological Services Commercial Insectary (Adelaide, Australia), were maintained under laboratory conditions until adult emergence. Adults of *A. melinus* were held in plastic cages (30 × 30 × 30 cm) covered with a fine mesh cloth. The wasp colony was reared on *A. aurantii*; on butternut squash (*Cucurbita moschata* Duchesne ex Lamarck) (26 °C, 40–60% relative humidity and L16: D8 h photoperiod) during their entire life. The original *A. aurantii* colony was established in 2016 from infested citrus fruit harvested in Western Australia (32.30° S, 116.01° E; 69 m AMSL). Fresh squash infested with scales (>1500 per squash) were placed in a cage with *A. melinus* adults which were released twice a week to maintain the *A. melinus* colony.

### 2.2. Plant Material

Three different citrus species (lemon, orange and mandarin) were used for behavioural assay and GC-MS analysis. Citrus fruit was collected from 25-year-old trees (spaced 4.5 × 5 m apart) on 15 September 2016, from Murdoch University citrus orchard located in Western Australia, same location as mentioned above. Non-infested or infested fruit from each citrus species was collected manually, stored in glass jars (10 cm diameter, 20 cm high) and transferred to laboratory within 1 h. The infested fruit samples were selected which had the most second and early third instar nymphs. Non-infested fruit samples were chosen to be free of any naturally physical damages as much as possible. Prior to test, the fruit were stored in the laboratory at 25 ± 2 °C and 45–55% RH for 5–7 days. The tested fruit samples were uniformed in weight, each sample consisting of one piece of fruit weighing about 150 g. The infested citrus fruit were selected with 300–400 s and early third instars of *A. aurantii*.

### 2.3. Extraction of VOCs and GC–MS Analysis

Random pieces of citrus fruit, either non-infested or infested, were placed individually into a 500 mL glass jar and sealed to equilibrate gases for 8, 16 and 20 h for mandarin, orange and lemon, respectively. The differences in the equilibration time for extracting VOCs for each fruit could be attributed to species characteristics and many other factors such as growing area, weather conditions and non-homogeneity of the fruit ripening stage. After this, VOCs were extracted by inserting and exposing a solid-phase microextraction (SPME) fibre with 50/30 µm Carboxen/DVB/PDMS (2 cm) (Sigma-Aldrich, Bellefonte, PA, USA) to the headspace of the glass jar over the fruit samples at room temperature at 25 ± 2 °C and 45–55% RH for 2 h. The fibre was then withdrawn into the needle and transferred to the injection port of the GC–MS system as described by [22]. The desorption time of the SPME fibre at GC-MS injector was 10 min [16]. Six replicates were conducted for each treatment. The VOCs were analysed by GC-MS using an Agilent 7820A GC (Mulgrave, Victoria, Australia), equipped with a DB-35ms (30 m × 0.25 mm × 0.25 µm) fused-silica capillary column (Agilent Technologies, Santa Clara, CA, USA), with a mass spectrometer detector 5977E (Agilent Technologies, USA) under splitless mode. The carrier gas was 99.99% helium supplied by BOC (Sydney, Australia). The operation conditions of GC-MS were 250 °C injector temperature; the oven temperature was initially programmed at 60 °C and increased to 270 °C (by 5 °C/min); the column flow rate and splitless were 20 mL/min at 1.5 min and 1:1; the total GC-MS run time was 45.4 min.

Compound peaks were deconvoluted by AMDIS version 2.72 and identified by searching the NIST 2014 MS database (US National Institute of Standards and Technology, Gaithersburg, MD, USA) with retention index confirmation. The experiment was repeated twice.

### 2.4. Y-Tube Olfactometer Behavioral Experiments

The olfactory response of *A. melinus* to host-associated cues was tested using a glass Y-tube olfactometer purchased from Volatile Collection Systems Co LLC (Gainesville, FL, USA). The Y-tube olfactometer design was similar to that described in [19] with slight modifications for the purpose of our research. Briefly, the Y-tube olfactometer consisted of a central tube (2.5 cm diameter, 15 cm long) and two lateral arms (2.5 cm diameter, 11 cm arm’s length) with ground glass fittings through which humidified air passed (0.5 mL/sec through each arm, controlled by up-stream flow meters). Each arm was connected to an extended glass tube (2 cm diameter, 6 cm long). The extended glass tube with the mesh barrier prevented the insects from escaping. The air was filtered with activated carbon traps, then passed through a chamber into each fruit, then VOCs and insects could be introduced. The air passed through the olfactometer and then the system was left to stabilise for 15 min prior to use.

Olfactometer studies were carried out at 25 ± 1 °C (45–55% RH) in a room illuminated with overhead daylight fluorescent tubes. A light was placed over the Y-tube olfactometer, and the surrounding area (around and below) was shrouded with white paper to block out any visual cues. *A. melinus* is an ectoparasitoid, so pupae could be isolated from host and host plant material without disrupting their development. From adult emergence until their use for bioassays, parasitoids were held in tubes (1 cm diameter, 4 cm length) and droplets of honey were provided throughout their lives. All wasps used for the bioassays were naive females without any previous contact with infested or non-infested host fruit. After placing parasitoids in the conditioning vial for a day, wasps were removed and placed in an empty container. On the day of bioassay, female wasps were isolated and held individually in glass vials with a drop of honey until tested. A total of 40 replicates (individual *A. melinus*) were conducted for each of the three citrus fruit species, and the four tested dosages of the synthetic VOCs. For each bioassay, a single female wasp was introduced into the central arm tube of the olfactometer. Once the wasp had reached the point where the arm divides (i.e., when the wasp reached a position to make a choice), a timer was started. If the parasitoid remained stationary for two minutes, it was considered as unresponsive. It was then excluded from the study and substituted with another one.

The olfactometer was reversed after half of the wasps in each replicate had been tested. After all ten parasitoids were assayed, the odour sources were removed, and the apparatus was cleaned with water and acetone, then dried and heated in an oven at 80 °C for at least half an hour between replicates. Overnight between treatments, new sources were introduced into the aeration chambers, and a further replicate of ten parasitoids was tested. For bioassays involving fruit, fresh and ripe fruit samples were tested to reduce the variation in volatile emissions between replicates.

To understand whether infested citrus fruit is attractive to *A. melinus*, at the first stage, mated females were allowed to choose between non-infested or infested fruit samples for each citrus species. A wasp was considered to have chosen a cue when it remained in the same chamber for at least 20 s, actively searching for a host. The bioassay replicate was considered complete when the wasp left the chamber. However, if wasps are unresponsive after two minutes in the system, they were eliminated from the study.

For assessment of the attraction of synthetic VOCs, two different concentrations (10 and 20 μL/mL) of d-limonene and *β*-ocimene (d-limonene purity 97%, 98% EE (GLC) Sigma-Aldrich, and *β*-ocimene purity ≥ 90% Toronto Research Chemicals, North York, ON, Canada) were dissolved in hexane. In each trial, 5 μL of solution was applied onto a filter paper (1.5 cm × 1.5 cm Whatman no. 1) and placed in an Erlenmeyer flask (250 mL) stoppered with a sealed adaptor. After solvent evaporation (20 sec), Erlenmeyer flask was connected to an arm of the Y-tube, and another arm received filter paper treated with 5 μL of pure hexane as a solvent control. Forty mated females were tested for each treatment.

For non-infested and infested citrus fruit bioassays, the female’s first choice, and search duration (time spent actively searching inside the arm) were recorded. For the bioassay of synthetic VOCs, only the female’s first choice was recorded. The activity recordings of parasitoids were used to assess the overall activity and the preference for each choice. The activity was measured by summing the number of entries into both arms by each wasp.

### 2.5. Statistical Analysis

To compare VOCs emissions between infested and non-infested fruit, the variance between peak areas was analysed for each compound and chemical class. Differences in volatile emissions between non-infested and infested fruit were analysed using non-parametric Mann-Whitney U test. Principal Component Analysis (PCA) was achieved on normalised values of each VOC to derive different variables (principal components) that summarise the original data. PCA analysis was performed using Metaboanalyst 3.0 (a comprehensive online tool suite for metabolomics data analysis). A likelihood chi-square test using Yates correction (with *p* = 0.05) for each choice-test was used to compare the proportion of parasitoids choosing a given cue. Time spent in the non-infested plant area and the infested plant area was compared using ANOVA, followed by Tukey’s post hoc tests. (SPSS version 24.0, SPSS, Inc., Chicago, IL, USA).

## 3. Results

### 3.1. Identification of VOCs

The data from GC-MS analysis showed that more than 34 different volatile compounds were separated and identified from the infested and non-infested citrus fruit and differential associated emissions attributed to herbivore activity were found for all fruit species (Table 1). The two naturally presented compounds in non-infested lemon, orange and mandarin, d-limonene and *β*-ocimene were significantly increased in the infested fruit (Table 1). However, *n*-hexadecanoic acid was exclusively produced by infested lemon fruit, and 5,4-di-*epi*-aristolochene was decreased after infestation, while 3,7-dimethyl-(*E*)-2,6-octadienal and α-bulnesene were only present in non-infested lemon and were not detected in infested lemon. There were no significant differences between the non-infested and infested mandarin fruits, but the compounds of γ-terpinene, alloocimene and alloaromadendrene were significantly increased in infested mandarins (Table 1). For orange, compounds were changed only in infested fruit, e.g., three more compounds, alloaromadendrene, 4,8-dimethyl-(3*E*)-1,3,7-nonatriene, and hexyl caproate were increased, and five compounds, acetic acid hexyl ester, alloocimene, α-tyerpineol, nerolidol and (3*E*,7*E*)-4,8,12-trimethyltrideca-1,3,7,11-tetraene were exclusively present in infested fruit (Table 1).

The PCA showed that the volatile compound profiles varied with infestation and citrus fruit species (Figure 1). d-Limonene and *β*-ocimene were the major VOCs released from all tested fruits. d-Limonene emitted 26.42%, 35.92% and 8.50% and *β*-ocimene emitted 32.83%, 31.51% and 17.46% of total volatile emissions from lemon, mandarin and orange fruit, respectively. Moreover, there were some major VOCs released from infested orange fruit such as alloaromadendrene, which accounted for 24.79% of total volatile emissions, and 19.65% and 23.48% of eudesma-4(14),7(11)-diene from lemon and orange fruit, respectively.

### 3.2. Behavioural Tests

The results of the Y-tube olfactory bioassays are presented in Figure 2. When olfactory cues were provided, *A. melinus* mated females showed significant preferences for lemon fruit infested with *A. aurantti* over non-infested ones, but no such preference was observed when compared between non-infested and infested oranges or mandarins. (lemon: χ^2^ = 4.900, df = 1, *p* = 0.027; orange: χ^2^ = 2.500, df = 1, *p* = 0.114; mandarin: χ^2^ = 1.600, df = 1, *p* = 0.206). There was no significant time differences of *A. melinus* to choose the chamber connected to *A. aurantti* infested citrus fruit. Indeed, *A. melinus* females slightly preferred and taken the initiative to the chambers containing infested fruit rather than health fruit (lemon: χ^2^ = 26.000, df = 21, *p* = 0.206; orange: χ^2^ =28.000, df = 21, *p* = 0.140; mandarin: χ^2^ = 26.333, df = 21, *p* = 0.194) (Table 2).

Mated *A. melinus* females, with no previous oviposition experience, were significantly attracted to the synthetic VOCs in all tests, with the exception of d-limonene at tested dosage 10 μL/mL (χ^2^ = 2500; df = 1; *p* = 0.114). However, mated *A. melinus* females preferred the reward of associated VOC more than hexane control in the case of d-limonene at the tested dosage of 20 μL/mL (χ^2^ = 7.410; df = 1; *p* = 0. 006), *β*-ocimene at the tested dosage 10 μL/mL (χ^2^ = 4.900, df = 1, *p* = 0.027) and *β*-ocimene at the tested dosage 20 μL/mL (χ^2^ = 12.100; df = 1; *p* = 0.001) (Figure 3).

## 4. Discussion

This report investigated the capability of an aphelinid parasitoid *A. melinus* to respond to citrus fruit chemical cues associated with damage induced by a phytophagous pest. The results indicated that *A. melinus* was able to discriminate between citrus fruit infested with *A. aurantii* and non-infested fruit by using olfactory cues.

Among 34 VOCs extracted and identified by SPME and GC-MS techniques, only two VOCs, d-limonene, and *β*-ocimene were increased in all infested three citrus species, which were already known as a constituent of the odours of citrus fruit and citrus oils [16,20,23]. However, since d-limonene and *β*-ocimene are attractive to many other parasitoid species [19,20,24], these two compounds can be considered as putative VOCs as attractants of *A. melinus*.

PCA analysis has highlighted that VOC emissions are particular for each species and chemical. Fruits differentially emitted the VOCs after insect infestation and the VOCs change depending on the citrus species. Indeed, several plants react to herbivore damage by producing blends of metabolites that change in number or proportion [25]. The increasing in VOCs follow the infestation could be explained for all species, but we also identified some particular compounds emitted under the condition of *A. aurantii* infestation. Besides d-limonene and *β*-ocimene, infested mandarin fruits displayed increased emissions of another three compounds (γ-terpinene, alloocimene and alloaromadendrene). Infested lemon fruits increased the emission of just d-limonene and *β*-ocimene and produced *n*-hexadecanoic acid. When attacked by *A. aurantii*, orange fruits similar to other species increased the production of d-limonene, *β*-ocimene, (3*E*)-4,8-dimethyl-,1,3,7-nonatriene, hexyl caproate and alloaromadendrene, and exclusively produced five compounds (acetic acid, hexyl ester, alloocimene, α-terpineol, nerolidol and (3E,7E)-4,8,12-trimethyltrideca-1,3,7,11-tetraene). Most of these compounds are known as common floral compounds, but interestingly some of them are also recognised as pheromones for several hymenopteran species [26].

Since the variance in VOCs depends mainly on the fruit species, we cannot exclude the possibility that other VOCs produced specifically by one species or increased in all species may act as an attractant toward *A. melinus*. Indeed, non-infested fruit also produce VOCs that are attractive to parasitoids [25], and aphelinid parasitoids, such as *A. melinus* could be able to perceive cues from non-infested plants to locate a host [19,20]. Therefore, short-range VOCs produced by the plants or as feeding larvae secretion could be useful for successful localisation [27]. Overall, since VOCs are known to act as attractive chemicals for several parasitoids [25,28,29], further research is needed to assess the activity of these compounds on parasitoid behaviour. Indeed, understanding tritrophic system communications has potential implications for biological control programs [30].

Olfactory cues from host-infested fruit are known to be essential for parasitoids during foraging for hosts, food and mates in complex environments [13,31]. For *A. melinus*, the source of volatile cues that the wasp uses to locate the host species were determined when those hosts occurred on more than one host plant species [19]. The evidence that olfactory cues from infested fruit evoke behavioural responses from mated *A. melinus* females, and the presence of chemicals that were produced exclusively or in higher amounts by infested citrus fruit, supported our hypothesis that VOCs could act as short-range attractants and played a key role during host-seeking. Biological assays conducted in the olfactometer indicated that *A. melinus* preferentially orientated towards the source of VOCs, that is, the presence of *A. aurantii* on the fruit is crucial for host location.

The VOCs d-limonene and *β*-ocimene have been reported as bouquet constituents of several citrus species [16,20]. d-Limonene and *β*-ocimene were identified as constituents of different infested fruit species that attracted other parasitoids such as *Agathis bishopi* and *Aphidius gifuensis* [24,32]. Many experiments have shown that, in the species examined, successful source location requires the presence of an attractive odour [33].

It is clear that *A. melinus* was not attracted to the pest when fruit material was absent, suggesting the importance of plant VOCs as long-range attractants of parasitoids, this result is consistent with that host insects do not always produce good cues [34]. This study confirmed that the aphelinid parasitoid *A. melinus* uses common VOCs to locate prey. The volatile profile analysis provides complementary information about the composition of the volatile chemicals potentially involved in the behaviours with 34 compounds varying during infestation by *A. aurantii*. Further electrophysiology and behavioural assays are expected to provide more accurate identification of the VOCs that contribute towards attracting *A. melinus* to infested host citrus fruit. Such information may prove useful for further studies aiming at developing semiochemical strategies that could be incorporated into an integrated management approach for enhancing existing pest control techniques for California red scale.

## 5. Conclusions

The aphelinid parasitoid *A. melinus* responded to citrus fruit chemical cues associated with damage induced by California red scale. The analytical data showed that more than 34 different volatile compounds were separated and identified from the infested and non-infested citrus species fruit. Interestingly, the three-species showed two common VOCs increased as a response of *A. aurantii* infestation: d-limonene and *β*-ocimene.

Olfactory cues seem to have a key role in host-seeking behaviour for the three citrus species. Synthetic VOCs may be useful to improve biological control programs especially after VOCs are recognized as VOCs of a number of parasitic wasps, but still, further studies are necessary to ensure the behavioural activity of this investigated VOCs toward *A. melinus* parasitoids under field condition.

## Figures and Tables

**Figure 1 insects-11-00044-f001:**
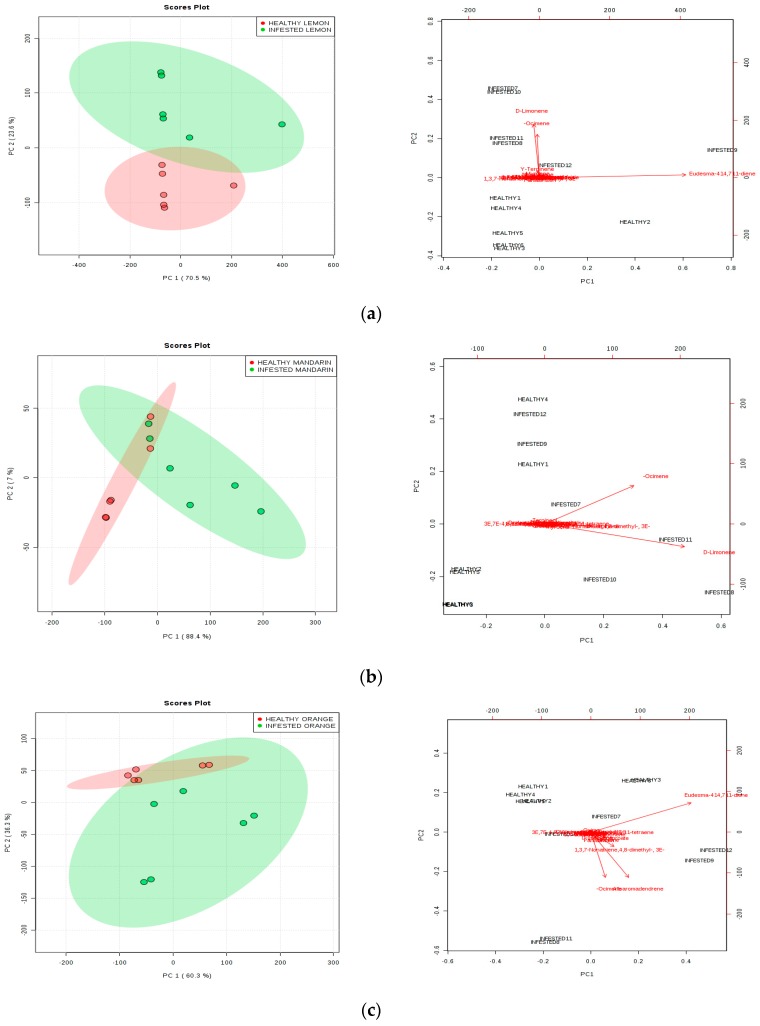
Principal Component Analysis (PCA) of volatile profiles from infested and non-infested fruit of three different citrus species. PCA scores plot and biplot of lemon (**a**), mandarin (**b**) and orange fruit (**c**), showing volatile correlations with the first and second principal component; PCA score plot, highlighting cluster of VOCs attributable to species or infestation status; PCA biplot highlighting changes in chemicals attribute to species or infestation status.

**Figure 2 insects-11-00044-f002:**
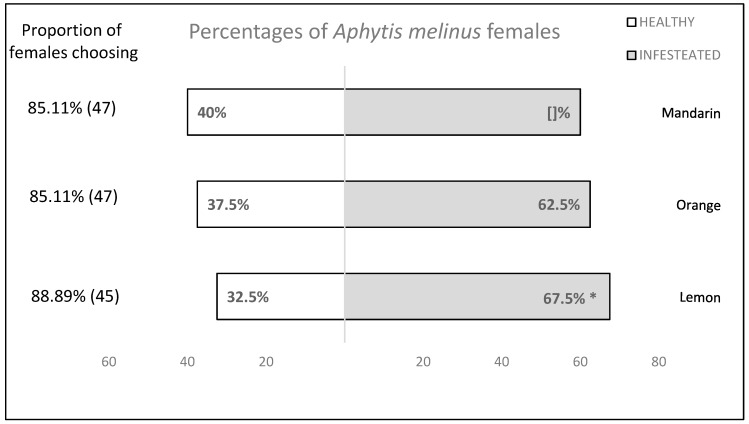
Attractiveness of *Aonidiella aurantii*-infested citrus fruit towards *Aphytis melinus* mated females. Two choice bioassays were conducted in a still air arena with citrus fruit, infested or not by California red scale, providing olfactory cues. Forty wasps were tested in each bioassay. For each test, asterisks indicate significant differences in the number of wasps choosing different cue (χ^2^ test with Yates correction, *p* < 0.05). To the left of the bars is the proportion of females that made a choice for one of the two odours, as well as the total number of females that were tested.

**Figure 3 insects-11-00044-f003:**
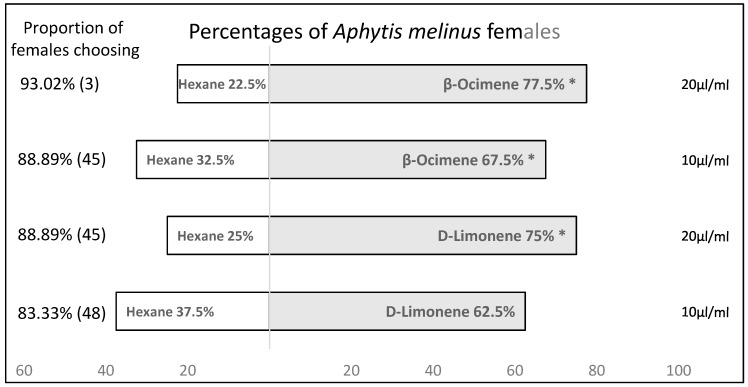
Attraction of female *Aphytis melinus* to VOCs differentially emitted by *A. aurantti*-infested citrus fruits. Choice bioassays were conducted in a Y-tube olfactometer presenting one of the two dosage levels (10 and 20 μL/mL) of citrus fruits VOCs vs. hexane. Asterisks indicate significant differences between numbers of VOC and control choices (χ^2^ test with Yates’ correction, *p* < 0.05). To the left of the bars is the proportion of females that made a choice for one of the two odours, as well as the total number of females that were tested.

**Table 1 insects-11-00044-t001:** Quantities of major volatile compounds released by non-infested and *Aonidiella aurantti* -infested citrus fruits through headspace sampling by SPME. (GC response—Peak Areas ± SE) × 10^6^.

Feature ID	NIST RI	Chemical Compounds	Lemon		Mandarin		Orange	
			Non-Infested	Infested	Non-Infested	Infested	Non-Infested	Infested
4.464–151.1	1302 *	Methoxyphenyl oxime	4.235 ± 1.382	4.749 ± 0.370	5.568 ± 1.504	4.512 ± 1.337	12.081 ± 2.677	9.811 ± 0.433
4.924–136.1	966	*β*-Thujene	n.d	n.d	0.731 ± 0.050	2.686 ± 1.127	n.d	n.d
5.921–136.1	979	*β*-Pinene	4.464 ± 0.064	5.269 ± 2.531	n.d	n.d	n.d	n.d
6.171–136.1	991	*β*-Myrcene	3.359 ± 2.009	6.115 ± 2.429	1.211 ± 0.405	6.752 ± 3.215	0.984 ± 0.571	1.636 ± 0.918
6.594–144.1	1011	3-Carene	n.d	n.d	n.d	n.d	n.d	2.754 ± 0.4046 *
7.043–136.1	1018	d-Limonene	30.852 ± 17.394	136.594 ± 18.874 *	31.585 ± 6.825	137.129 ± 30.916 *	24.716 ± 1.878	40.918 ± 2.773 *
7.429–136.1	1037	*β*-Ocimene	67.916 ± 6.775	169.708 ± 13.934 *	45.039 ± 18.482	120.268 ± 10.677 *	20.812 ± 3.123	83.942 ± 15.674 *
7.625–136.1	1047	γ-Terpinene	3.686 ± 2.522	12.214 ± 7.216	7.117 ± 2.606	41.078 ± 11.193 *	0.573 ± 0.498	1.633 ± 0.929
8.258–136.1	1088	Terpinolene	2.449 ± 1.443	1.988 ± 1.018	1.431 ± 0.251	5.572 ± 2.170	n.d	n.d
8.808–150.1	1116	(3*E*)-4,8-Dimethyl-1,3,7-nonatriene	35.130 ± 15.029	18.611 ± 1.789	17.214 ± 7.774	31.584 ± 16.498	3.774 ± 1.832	32.767 ± 10.184 *
9.092–134.1	1131	Cosmene	4.025 ± 0.683	6.200 ± 0.870	0.635 ± 0.319	1.238 ± 0.622	1.561 ± 0.586	3.577 ± 1.367
9.338–136.1	1144	Alloocimene	3.653 ± 2.236	5.718 ± 2.126	1.108 ± 0.198	6.906 ± 0.021 *	n.d	2.409 ± 0.641 *
9.867–128.2	1177	*trans*-Isopulegone	n.d	n.d	0.191 ± 0.191	0.218 ± 0.218	n.d	n.d
10.086–154.1	1182	4-Terpineol	n.d	n.d	0.284 ± 0.146	0.896 ± 0.658	n.d	n.d
10.309–154.1	1189	α-Terpineol	1.081 ± 0.567	1.815 ± 1.334	0.525 ± 0.264	2.975 ± 2.476	n.d	3.004 ± 1.176 *
10.488–170.2	1200	Dodecane	n.d	n.d	0.785 ± 0.119	1.126 ± 0.195	n.d	n.d
10.614–156.1	1206	Decanal	n.d	n.d	1.272 ± 0.642	1.312 ± 0.667	1.416 ± 0.302	1.438 ± 0.209
11.943–152.1	1270	3,7-Dimethyl-2,6-octadienal	0.753 ± 0.386	n.d	n.d	n.d	n.d	n.d
12.478–212.2	1275	2,6,11-Trimethyldodecane	1.369 ± 0.819	0.745 ± 0.373	1.196 ± 0.051	1.648 ± 0.335	1.371 ± 0.108	1.766 ± 0.213
14.111–200.3	1384	Hexyl caproate	n.d	n.d	n.d	n.d	4.375 ± 1.713	18.305 ± 6.018 *
14.206–204.2	1398	*β*-Elemene	7.433 ± 2.337	5.899 ± 0.894	n.d	n.d	10.546 ± 4.774	20.594 ± 5.217
14.922–204.2	1461	Alloaromadendrene	17.374 ± 5.366	13.587 ± 0.729	3.454 ± 0.358	7.713 ± 0.894 *	39.032 ± 7.499	119.291 ± 13.459 *
15.236–204.2	1490	α-Bulnesene	1.309 ± 0.805	n.d	n.d	n.d	n.d	n.d
15.384–204.2	1440	Aromandendrene	n.d	n.d	n.d	n.d	4.920 ± 1.749	9.635 ± 2.296
15.784–204.2	1469	5,4-di-*epi*-Aristolochene	2.391 ± 0.345 *	1.435 ± 0.103	n.d	n.d	3.789 ± 1.474	7.687 ± 0.861
15.941–204.2	1527	Panasinsene	16.519 ± 6.276	10.936 ± 2.061	n.d	n.d	11.113 ± 4.743	20.255 ± 2.813
16.283–204.2	1544	Eudesma-4(14),7(11)-diene	51.037 ± 45.977	101.571 ± 76.637	n.d	n.d	100.882 ± 27.236	113.043 ± 38.383
17.039–204.2	1556	Guaia-3,9-diene	1.705 ± 0.296	2.091 ± 0.595	n.d	n.d	n.d	n.d
17.323–222.2	1564	Nerolidol	3.129 ± 2.278	2.652 ± 1.351	n.d	n.d	n.d	1.247 ± 0.644
17.532–218.2	1581	(3*E*,7*E*)-4,8,12-Trimethyltrideca-1,3,7,11- tetraene	n.d	n.d	1.941 ± 0.766	2.587 ± 1.083	n.d	1.582 ± 0.821 *
17.849–220.2	1586	*trans*-(Z)-*α*-Bisabolene epoxide	1.355 ± 0.319	1.322 ± 0.359	0.142 ± 0.1416	0.155 ± 0.155	n.d	n.d
18.945–222.2	1660	Neointermedeol	3.218 ± 0.332	4.672 ± 0.699	0.348 ± 0.182	0.566 ± 0.288	1.275 ± 0.332	2.305±.389
23.368–256.2	1968	n-Hexadecanoic acid	n.d	0.920 ± 0.920	3.098 ± 1.076	3.336 ± 0.928	n.d	n.d
44.331–722.6	4932 *	Trimyristin	2.843 ± 2.201	2.088 ± 0.529	1.343 ± 0.458	1.416 ± 0.266	n.d	n.d

Asterisks indicate significant differences between uninfected and infested fruit at *p* < 0.05; Feature ID includes retention time (min) and *m/z* ratio *z*; RI retention index obtained from the NIST database; * Estimated non-polar retention index (*n*-alkane NIST scale); SE = standard error; n.d = not detected. Each value represents the peak area (mean ± SE) of *n* = 6 analyses.

**Table 2 insects-11-00044-t002:** Choice time spent by *Aphytis melinus* females during searching behaviour on non-infested and *Aonidiella aurantii* -infested citrus fruits in Y-tube Olfactometry.

Species	Infested Citrus Fruit		Non-Infested Citrus Fruit		F	*p*-Value
	Choice Time (s) Mean ± SE	Replicates	Choice Time (s) Mean ± SE	Replicates		
Lemon	148.15 ± 13.482	27	133.31 ± 14.604	13	0.558	0.206 ^ns^
Orange	153.40 ± 10.523	25	119.13 ± 12.084	15	4.573	0.140 ^ns^
Mandarin	139.53 ± 12.802	23	144.06 ± 15.412	17	0.051	0.194 ^ns^

ns: no significant differences in choice time spent by *Aphytis melinus* between uninfected and infested fruit at *p* < 0.05; ns: not significant.

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
