# Peer review of "Evaluation of d-Limonene and β-Ocimene as Attractants of Aphytis melinus (Hymenoptera: Aphelinidae), a Parasitoid of Aonidiella aurantii (Hemiptera: Diaspididae) on Citrus spp."

_insects, 2020, doi:10.3390/insects11010044_

Round 1
Reviewer 1 Report
The present paper is interesting, even though the novelty is somewhat limited. Overall the study is scientifically interesting but suffers from some structural and language problems, that are described below in more detail. Besides, some experimental information needs to be clarified.
Some general remarks are the following:
The author referred to citrus healthy fruits in several diverse ways. “healthy”, “non-infested”, “uninfested”. I recommend the same pattern should be used throughout the manuscript. Please standardize throughout the entire manuscript. I suggest "non-infested citrus fruit".
Please change throughout the manuscript “ml” and “μl” in “mL” and “μL”.
In Table 1 and throughout the manuscript follow IUPAC rules for the nomenclature of VOCs or their common names.
For example:
“Cyclohexene, 1-methyl-4-(1-methylethylidene)-” must be written either “1-methyl-4-(1-methylethylidene)-1-cyclohexene” or “terpinolene”.
“1,6,10-Dodecatrien-3-ol,3,7,11-trimethyl-” must be written either “3,7,11-trimethyl-1,6,10-dodecatrien-3-ol” or “nerolidol”.
Abstract
Line 18: delete “as plant-fed herbivores” or explain it.
Line 25: I suggest deleting “putative”.
Introduction
Line 43: add Osbeck after Citrus limon (L.), add “x” before paradisi Macf.
Line 44: please define that there are two different types of mandarin.
Line 45: add “A. aurantii” after scale.
Line 56: Change “herbivorous-induced plant volatiles” in “herbivore-induced plant volatiles” throughout the manuscript and add here the appropriate abbreviation.
Materials and Methods
Line 92: delete limon.
Line 102: The authors must clarify why the equilibration time was different for each fruit.
Line 103: put a comma after “this”.
Lines 109-111 refer to plant material please remove them from here. They better fit after line 99.
Line 113: specify length, i.d., film thickness. It is more appropriate to write 0.25 mm instead of 250 μm.
Line 121: please clarify “The experiment was repeated twice”.
Line 122: I recommend another title like “Y-Tube Olfactometer Behavioral Experiments/ Tests” Please incorporate in the manuscript the number of replicates for the parasitoids in all behavioral experiments.
Lines 137-138: please change in “held in tubes (1cm diameter, 4cm length)”.
Line 142: The “glass vials” were different from tubes of line 138?
Line 175: I do not think that ANOVA is the appropriate procedure. I would suggest the non-parametric Mann-Whitney U test.
Line 179: “α” I think should be “p”
Results
Firstly, there is a contradiction in Table 1 and Figure 1. Authors stated in MM section that six replicates were conducted for each treatment. In Table 1, the authors stated that n=6. However, in Figure 1, there are only three replicates per treatment. Please clarify.
Lines 201-205: Since the statistical analysis of VOCs was based on the volatile peak area, why authors present the percentage of volatiles? Results that emerged from PCA biplots are not presented.
The number of replicates and the number of individuals with no choice should be included in figures. Do not use negative numbers in axis, please correct.
In Table 1, please define if retention indices are calculated or they are literature data. How RIs were calculated? If they are calculated, please provide also literature RI.
Discussion
My main criticism is about the discussion, I think that results are interesting but should be better discussed.
The authors state, at the beginning (lines 252-253) “The results indicated that A. melinus was able to discriminate between citrus fruit infested with A. aurantii and non-infested fruit by using olfactory cues”, and in the conclusions (line 301) “The aphelinid parasitoid A. melinus responded to citrus fruit chemical cues..” but in my opinion the Y-tube olfactometer do not supports such a strong statement for all citrus fruit. Only lemon infested with A. aurantii was significantly attracted by A. melinus mated females.
Lines 255 and 303: 34 or 35 VOCs. Please, correct.
Line 266: Change “Y” in “γ”.
Reviewer 2 Report
The authors of “Evaluation of D-Limonene and ?-Ocimene as 2 attractants of Aphytis melinus (Hymenoptera: 3 Aphelinidae), a parasitoid of Aonidiella aurantii 4 (Hemiptera: Diaspididae) on Citrus spp.” studied with three Citrus species the attraction of parasitoids to herbivore-damaged fruits. They found out that VOCs emitted from fruits differ between species and between herbivore-damaged or intact fruits. The parasitoids were more attracted to the infested fruits, and also individual VOCs of infested fruits. So, the parasitoids could distinguish infected fruits from uninfected ones, and the parasitoids could be used in biological control. The experiment was well planned and conducted. The data is analysed well, and the text is rather understandable. The text could become more concise with some copy-editing.
Throughout the manuscript you use various terms to explain the same thing: volatiles, VOCs, HIPVs, herbivore-induced volatiles, kairomones. This can be confusing for the readers. Pick one term and stick to that, preferable either VOCs or HIPVs.
Picking one term is also for the study species. At first mention tell both English and scientific names but then stick to either one. Check also the spelling of the scientific name in each instance. There are now some typos.
Here are some more detailed comments:
lines 19-21: Tell clearly here to the readers who are not that familiar with the species that which one is the herbivore, and which one the parasitoid. This is told in the title, but would be useful to repeat here for clarity.
line 23: you tell what VOCs means already on line 17, so no need to repeat that here
line 31: What was then the other dosage?
lines 39-40: Rephrase this sentence, now it’s rather unclear.
line 86: I was surprised that here you used as host plant butternut squash. Was it because then the parasitoids would not learn the specific odours of the Citrus trees?
lines 88-89: correct to: 32.30°S, 116.01°E; 69 m AMSL
line 102: Why these different times for the species? Explain a bit more this.
lines 145-146: How many parasitoids were excluded due to not making a choice?
line 191: ininfested?
Figure 1: Change the colours used in this figure. Now colour-blind people (up to 5 % of people) cannot distinguish the red and green used here.
lines 243-244: The table has no column of RT.
Table 2: This information would be much easier to understand as a figure.
line 261: What do you mean with “they” here?
Author Response
Throughout the manuscript you use various terms to explain the same thing: volatiles, VOCs, HIPVs, herbivore-induced volatiles, kairomones. This can be confusing for the readers. Pick one term and stick to that, preferable either VOCs or HIPVsAll “volatiles, HIPVs, herbivore-induced volatiles, kairomones” have been changed to “VOCs” throughout the manuscript.
Picking one term is also for the study species. At first mention tell both English and scientific names but then stick to either one. Check also the spelling of the scientific name in each instance. There are now some typos.
Changed accordingly, thanks!
Here are some more detailed comments:
lines 19-21: Tell clearly here to the readers who are not that familiar with the species that which one is the herbivore, and which one the parasitoid. This is told in the title, but would be useful to repeat here for clarity.We have mentioned here to the A. melinus as a parasitoid to distinguish it from CRS to readers who are not familiar with the species.
line 23: you tell what VOCs means already on line 17, so no need to repeat that hereLine 17, we have talked in general about the role of VOCs as chemical signals used by parasitoids during hosting location, while line 23, we have mentioned how many VOCs have been identified from three species of citrus fruit using via solid-phase microextraction (SPME) coupled with gas chromatography-mass spectrometry (GC-MS), therefore, here no repeated contents.
line 31: What was then the other dosage?Here indicated that all doses used have a significant effect on attracting A. melinus, except for the dose of 10µL\Ml of D-limonene, which means that the parasitoids showed positive chemotaxis toward the dosage of 10µL\Ml of B-Ocimene.
lines 39-40: Rephrase this sentence, now it’s rather unclear.The sentence has been paraphrased.
line 86: I was surprised that here you used as host plant butternut squash. Was it because then the parasitoids would not learn the specific odours of the Citrus trees?Parasitoid production was carried out following standard commercial procedures using an alternative host plant for California red scale, butternut squash, due to its ability to last for a long time.
lines 88-89: correct to: 30°S, 116.01°E; 69 m AMSLChanged “32.30°s 116.01°e 69m AMSL” to “32.30°S, 116.01°E; 69 m AMSL”.
line 102: Why these different times for the species? Explain a bit more this.Our previous published results (Journal of Biosciences and Medicines 5(03), pp.176-186 “by Mohammed, K., Agarwal, M., Newman, J. and Ren, Y., 2017) have optimized the best conditions to find out an optimum method for extracting volatile organic compounds (VOCs) which contribute to the aroma of different species of citrus fruit (orange, lemon, lime, and mandarin).Based on optimization of headspace solid-phase microextraction conditions for the identification of volatiles compounds from the whole fruit of lemon, lime, mandarin and orange, we have applied the equilibration time for each species in current experiments to get the highest quality peak areas and quantity of extracted volatiles. The differences in the equilibration time for each fruit are generally attributed to species characteristics and many other factors such as growing area, weather conditions and non-homogeneity of the fruit ripening stage.
lines 145-146: How many parasitoids were excluded due to not making a choice?Just a few parasitoids were excluded from the study due to not making a choice and substituted with another one.
line 191: ininfested?It has been corrected.
Figure 1: Change the colours used in this figure. Now colour-blind people (up to 5 % of people) cannot distinguish the red and green used here.The statistical program (Metaboanalyst 3.0) autometiclly generate and manage the colours, therefore, the colours can’t be changed.
lines 243-244: The table has no column of RT.“RT is retention times” has been cleared because RT is defined in “Feature ID”.
Table 2: This information would be much easier to understand as a figure.We completely agree with reviewer’s comments, but there is some information that will be difficult to explain such as the number of replicates if the table converted into a figure.
line 261: What do you mean with “they” here?Changed “they” to “the VOCs”.
Please see the attachment

This manuscript is a resubmission of an earlier submission. The following is a list of the peer review reports and author responses from that submission.